# Systemic Infection by *Angiostrongylus vasorum* in a Fennec (*Vulpes*
*zerda*) in an Italian Zoological Garden

**DOI:** 10.3390/pathogens11090965

**Published:** 2022-08-24

**Authors:** Valentina Galietta, Claudia Eleni, Caterina Raso, Cristiano Cocumelli, Klaus G. Friedrich, Pilar Di Cerbo, Manuela Iurescia, Elena L. Diaconu, Patricia Alba, Claudio De Liberato

**Affiliations:** 1Istituto Zooprofilattico Sperimentale del Lazio e della Toscana “M. Aleandri”, Via Appia Nuova 1411, 00178 Rome, Italy; 2Fondazione Bioparco, Viale del Giardino Zoologico 20, 00197 Rome, Italy

**Keywords:** *Angiostrongylus vasorum*, Canidae, *Vulpes zerda*, parasites, zoo, Italy

## Abstract

This paper reported a case of a metastrongyloid nematode *Angiostrongylus vasorum* infection in a fennec (*Vulpes zerda*) kept in a zoo in central Italy. The fennec had shown paralysis of the hind limbs, anorexia, weakness and respiratory signs before death. Cardiomegaly and granulomatous pneumonia were the major anatomopathological findings. Inflammatory lesions associated with parasitic larvae were observed in the lungs, brain, liver, heart, spinal cord and kidney of the fennec at histology. *A. vasorum* diagnosis was confirmed by both morphological and molecular identification of adult worms recovered at necropsy. Fennecs are active predators and maintain their hunting behaviour in captivity. Hence, it is likely that the animal was exposed to infection by preying on parasitised gastropods, intermediate hosts of *A. vasorum*, entering zoo enclosures from the surrounding environment. This is the first report of *A. vasorum* systemic infection in a captive fennec (*V. zerda)* in a zoo in Italy.

## 1. Introduction

*Angiostrongylus vasorum* (Nematoda, Metastrongyloidea) is a parasitic nematode residing in the right side of the heart and pulmonary arteries of many canid species [1]. Gastropod molluscs, slugs and snails, of many different families and genera, are its intermediate hosts [2]. The red fox (*Vulpes vulpes*) is its reservoir, and high prevalence of *A. vasorum* has been reported in populations of this species in many countries all over the world [3]. Other wild canids are natural hosts of *A. vasorum*, such as the grey wolf (*Canis lupus*), golden jackal (*Canis aureus*) and coyote (*Canis latrans*) [3,4,5]. *Angiostrongylus vasorum* causes cardiopulmonary disease in many species such as dogs, red foxes, red pandas and Eurasian badgers [6,7,8,9,10,11]. In dogs, *A. vasorum* can cause either a subclinical infection or a wide range of clinical outcomes, including mild-to-severe respiratory disease and bleeding disorders [3]. Respiratory signs are mainly characterised by cough, tachypnoea and dyspnoea. These signs are caused by verminous pneumonia provoked by eggs and first-stage larvae (L1) [12]. In the most severe forms, pulmonary hypertension may occur due to an increased vascular resistance induced by thromboembolic phenomena and extensive inflammatory processes involving pulmonary vessels and parenchyma caused by the presence of the larvae [13,14]. Although more rarely, angiostrongylosis can cause severe neurological signs, such as ataxia, paresis and paralysis [15]. Neurological signs are caused by haemorrhage, resulting from the parasite-induced coagulation disorder [16], and by damage caused by aberrantly migrating larvae [17]. The most severe forms of angiostrongylosis, mainly observed in younger animals, can be life-threatening [3,12,18,19,20]. Histopathological findings in diseased canids include granulomatous inflammation and interstitial fibrosis of the lung and thrombosis of pulmonary arteries, with fibrin and parasites [20,21]. Cases of disseminated angiostrongylosis have been reported in dogs and red foxes, with granulomas associated with nematode eggs and larvae in the tracheobronchial lymph nodes, brain, kidney, spleen, liver, eye and adrenal gland [10,11,21,22,23,24]. In these cases, due to the widespread localisation of the parasites, signs can be very diverse.

Until the 1990s, *A. vasorum* distribution in Europe was known to be highly localised in those that were considered “hotspots” of circulation [25,26]. Since then, the parasite apparently spread to new areas, with records in many European countries from areas in which parasite presence was not previously recorded [1,26]. Concurrently, it emerged as an increasing clinical concern in dogs both in endemic and new areas [26,27,28,29].

The spread of *A. vasorum* and establishment of further new endemic foci were also reported from Italy. By 2002, angiostrongylosis was reported with increasing frequency in dogs, and now it is considered endemic throughout the country [30]. Moreover, studies on its sylvatic hosts, foxes and wolves, highlighted its spread in the country [4,11,31,32].

*Angiostrongylus vasorum* can be a threat also for animals kept in captivity in zoos and collections. Cases of infection and even fatalities have been reported in captive red pandas (*Ailurus fulgens*) in European zoos [10,33], and the congeneric species *Angiostrongylus dujardini* caused the death of suricates (*Suricata suricatta*) and callitrichid monkeys (*Saguinus oedipus* and *Callimico goeldii*) in a zoo in central Italy (Tuscany region) [34].

Hence, it is evident that *A. vasorum* and *Angiostrongylus* spp., in general, are able to be introduced in zoos presumably within gastropod intermediate hosts entering the enclosures. Thus, captive animals in zoos are at risk of life-threatening angiostrongylosis. The present report described for the first time a severe infection due to *A. vasorum* in a captive fennec (*Vulpes zerda*), kept in a zoo in central Italy, such that the zoo veterinary personnel had to euthanise the animal. 

## 2. Case Presentation

The case occurred in the Bioparco zoological garden, Rome (Latium region, Italy). The Bioparco is one of the oldest zoological gardens in Europe, founded in 1911. It is located in the city centre, covering an area of 18 ha and housing about 1000 specimens belonging to almost 200 species of mammals, birds and reptiles. It was recently renovated in accordance with the new concept of “Zoo without bars” to improve animal welfare. Animals live in large spaces with the reconstruction of the natural habitats suitable for each species. To avoid contact with people, glass screens or ditches border animal cages and pens.

The fennec, a 14 month-old female species born at Bioparco, used to live in a semi-natural pen with other five conspecific individuals and was brought to the Istituto Zooprofilattico Sperimentale del Lazio e della Toscana “M. Aleandri” (IZSLT) for post mortem examination after being euthanised. One month before death, it had presented weakness, ataxia and paraparesis. Haemochromocytometric tests revealed mild lymphocytosis. It was treated with a corticosteroid (prednisolone, 1 mg/kg/die) and antibiotic therapy (Enrofloxacin, 5 mg/kg/die) for 3 weeks. Respiratory signs, such as cough and severe dyspnoea, emerged 2 days before death. After the appearance of these last signs, thoracic radiograph was performed, showing a diffuse nodular broncho-interstitial lung pattern (Figure 1). Due to serious clinical condition and lack of improvement with the treatment protocol, it was decided to carry out euthanasia.

Post mortem examination revealed poor body conditions (Figure 2) and pale mucous membranes. Major gross lesions were found in the lungs, where large, congested areas of consolidation, particularly located in the caudal lobes, were observed. At the section of lung parenchyma, necrotic and necrotic–haemorrhagic areas were found, associated with adult worms protruding from the lumen of the pulmonary arteries. Numerous adult worms were also detected in the right heart ventricle. Cardiomegaly was observed, with right ventricular hypertrophy associated with atrial dilatation. Catarrhal gastritis with haemorrhagic petechiae on the mucosa was seen. The kidneys appeared pale, with whitish streaks that deepened into the cortex. Numerous petechial haemorrhages were also found on the cerebral meninges.

Samples of the lung, heart, liver, spleen, kidney, brain, thoracolumbar spinal cord and intestine were collected, fixed in 10% buffered formalin and then routinely processed for histopathological examination.

Microscopically, lungs showed severe chronic granulomatous pneumonia, associated to the presence of eggs and larvae in the granulomatous foci. Vascular congestion and disseminated alveolar haemorrhages were found. In the lumen of some pulmonary arterioles, thrombotic formations with adult worms were seen (Figure 3a). Multifocal lymphoplasmacytic infiltrates and granulomatous foci with nematode larvae were observed in the myocardium of the right ventricle, spleen, intestinal submucosa and portal areas of the liver. An adult worm was also found in a branch of the portal vein (Figure 3b). Extensive interstitial lymphoplasmacytic nephritis associated with marked fibrosis and scattered granulomas containing nematode larvae were seen in the kidneys; larvae were also observed in many renal glomeruli (Figure 3c).

Microscopic examination of the brain revealed a mild, multifocal, granulomatous meningoencephalitis with the presence of nematode larvae in several small cerebral vessels, lymphoplasmacytic perivascular inflammation and disseminated microhaemorrhages in meninges and cerebral parenchyma (Figure 3d). Similar findings were found in the lumbar spinal cord (Figure 3e).

The first-stage larvae of *A. vasorum* were microscopically observed in the impression smears of the lungs (Figure 4). Adult nematodes retrieved in the right heart ventricle were washed and stored in 70% ethanol until morphological and molecular identification. Two adult female species were identified as *A. vasorum*, based on the size and typical morphology of ovaries, wrapped in a coil around the intestine [2]. Their identification was subsequently confirmed by individual genetic analysis.

For molecular analysis, parasites were incubated at 56 °C in ATL and proteinase K until complete lysis. Lysate was treated with RNase A at 37 °C. The total DNA of the nematode was extracted using a DNeasy Blood & Tissue kit (Qiagen, Hilden, Germany) following the manufacturer’s protocol. Whole DNA was subjected to a PCR assay for the amplification of the internal transcribed spacer 2 (ITS-2) rDNA locus using multiple forward (NC1) and reverse (NC2) primers mixed in equal proportions as previously described (https://www.nemabiome.ca/ accessed on 1 October 2021) [35] with slight modifications. PCR products were purified with AMPure XP Magnetic Beads (1X) (Beckman Coulter, Inc, Brea, CA, USA), and an Illumina index was added to the amplicon using PCR amplification. 

Sequencing was performed on an Illumina MiSeq platform. Raw reads were cleaned by using Cutadapt version 3.4 (accessed from IZSLT, Rome, Italy) [36], which finds and removes adapter sequences, primers, poly-A tails and other types of unwanted sequence. The cleaned reads were then analysed by using DADA2 (R package version 1.12.1 accessed from IZSLT, Rome, Italy) [37], which implements a complete pipeline to turn paired-end fastq files from the sequencer into merged, denoised, chimera-free, inferred sample sequences. The final length of the nematode amplicons was in the range of 250–300 bp, and a consensus of 260 bp was obtained. 

Identification of the amplicon consensus by comparing it with the nucleotide database (nr/nt; NCBI), using BLASTN online, indicated that the sequence had a 99.6% identity (100% coverage) with the ITS sequence of the *Angiostrongylus vasorum* isolate D15-3601 internal transcribed spacer (MT345058.1). The consensus obtained has been deposited in the European Nucleotide Archive (ENA) at EMBL-EBI under accession number OX249874.

Following the report on the *A. vasorum* infection in the euthanised fennec, other individuals of the same species living in the same pen were tested for the presence of this parasite via the Baerman technique, considered the gold standard for the diagnosis of broncho-pulmonary strongyles. Each animal was tested twice. No other cases of infection were highlighted.

## 3. Discussion

To the best of our knowledge, this is the first report of *A. vasorum* causing very severe symptoms in an animal kept in a zoo in Italy, such that veterinarians had to euthanise the animal, and it is the first report in *V. zerda*. In this respect, the clinical picture and pathological findings observed in the fennec suggest that the parasite was presumably leading the animal to death. Remarkably, the animal of our study displayed a severe form of systemic angiostrongylosis, with multiorgan involvement. However, the animal only showed major neurological signs, while respiratory signs occurred only a few days before death. For this reason, the parasitic infection was not initially suspected, and parasitological exams were not performed. The severity and presentation of the disease could have been influenced by individual health conditions, age and immune status, all factors that can have a role in the progression of the parasitoses, as previously reported in other species [38]. Therefore, given the clinical picture of the fennec, *A. vasorum* should be considered among the differential diagnoses not only in case of respiratory signs but also with neurological disorders.

The fennec was born at Bioparco less than 1 year before death; hence, infection was acquired in the zoo enclosure. The most probable mean of introduction of *A. vasorum* in a zoo is through gastropods. Considering the metastrongyloid life cycle, infections in zoos are usually believed to be caused by the ingestion of infected slugs and snails entering the enclosures [10,39,40]. Fennecs are active predators of different kinds of small animals, and they do not lose their hunting behaviour when in captivity. The Bioparco of Rome is located within Villa Borghese, a historical villa in the city centre, where every day many people take their dogs for a walk. Interestingly, the fennecs’ enclosure is just one hundred metres away from the area of Villa Borghese specifically devoted to dogs’ leisure (known as the “dogs valley”) and therefore characterised by a high dog fecalisation. In the presented case, the most likely way of infection was the captive animal preying on a gastropod that arrived from the neighbouring area of Villa Borghese and harbouring parasite larvae.

In conclusion, *A. vasorum* should be considered among the parasites reported to be introduced into zoo enclosures, sometimes leading to captive animals’ death [10,32]. Therefore, it would be advisable to improve the methods for the prevention and early diagnosis of these parasites in zoos.

## Figures and Tables

**Figure 1 pathogens-11-00965-f001:**
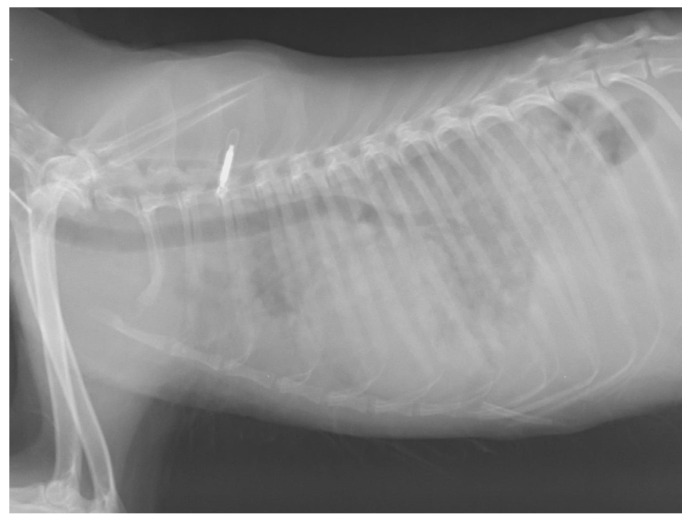
Thoracic radiograph of the fennec. Image shows lateral radiographic projection of the chest in which a diffuse nodular broncho-interstitial lung pattern can be observed.

**Figure 2 pathogens-11-00965-f002:**
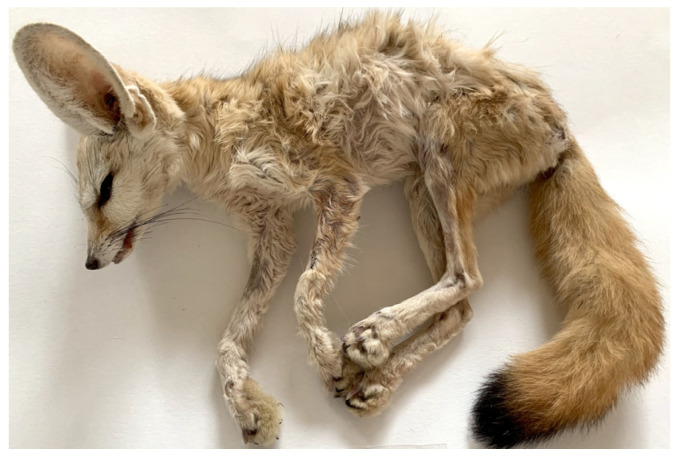
Fennec parasitised by *Angiostrongylus vasorum*. The image, taken at necropsy, shows the severe cachectic condition of the animal.

**Figure 3 pathogens-11-00965-f003:**
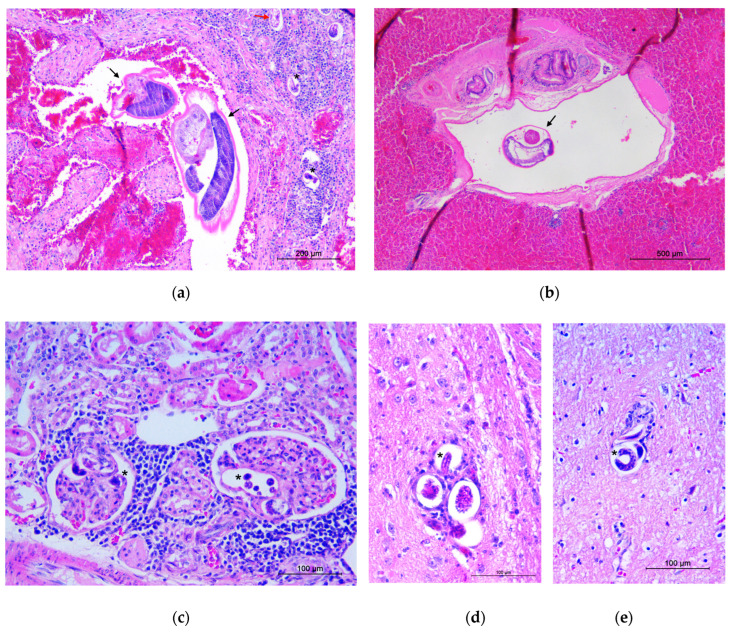
Detection of *Angiostrongylus vasorum* in parasitised organs. (**a**) Section of lung: an adult parasite in the pulmonary artery (black arrows), egg (red arrow) and larvae in granulomatous foci of the surrounding parenchyma (asterisks). (**b**) Section of liver: an adult worm (arrow) can be observed in a branch of the portal vein. (**c**) Section of kidney: larvae in glomerular capillaries, associated with periglomerular lymphoplasmacytic nephritis (asterisks). (**d**) Section of brain: larvae in the small cerebral vessels (asterisk) with mild lymphoplasmacytic perivascular inflammation. (**e**) Section of lumbar spinal cord: larvae in small spinal cord vessels (asterisk).

**Figure 4 pathogens-11-00965-f004:**
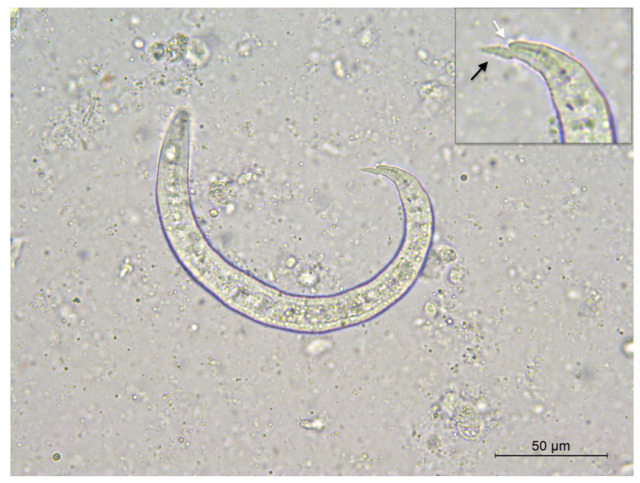
*Angiostrongylus vasorum* first-stage larva in impression smears of lungs. Note the pointed and slightly sinusoidal tail (black arrow) with dorsal indentation (white arrow).

## Data Availability

All data referred to in this study are available in the manuscript.

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
