# Peer review of "Systemic Infection by Angiostrongylus vasorum in a Fennec (Vulpes zerda) in an Italian Zoological Garden"

_pathogens, 2022, doi:10.3390/pathogens11090965_

Round 1
Author Response
Ms presents the detail description of disseminated Angiostrongylus vasorum infection in a fennec from Zoological garden. Nematode infection resulted with the death of the animal. I wonder if this case presentation should better fit a journal dedicated to veterinary science.
Major comment
The lack of detail data on molecular identification: in case of genotyped parasite, the obtained sequence shall be deposited in public database, i.e. GenBAnk with certain accession number. Furthermore, the authors shall present the details of ‘best match’ from the database: including the length of the compared sequence, the % of identity to certain sequence, the origin of that sequence (host, country). Currently no such data is provided.
- We thank the reviewer for his/her request. In the revised version, we have provided a more detailed description of molecular analysis methods and results. The results of sequencing have been deposited in database and accession number is now provided.
Minor comments
Please expand ‘A.’ to Angiostrongylus at the beginning of sentence.
- Done
Please provide full common names of animals: rgey wolf, red fox, golden jackal, etc.
- Done
Lines 49-53: please cite the recent review by Fuehrer et al. 2021
- We appreciate the reviewer's suggestion. We have also included this recent review among the references.
Line 62: where was it exactly in Italy? Similar region?
- The cases, cited in line 62, came from a zoo in central Italy, more precisely in the Tuscany region, which borders the Latium region, where we found the case of fennec angiostrongylosis, describing in the paper. Following the comment, we have indicated in detail the origin of the cases mentioned.
Lines 142-144: not clear description; do you mean PCR amplification and sequencing of selected marker/gene?
- The paragraph has been rephrased and more details about the procedure have been provided.
Lines 151-152: first major comment – lack of relevant details
- We have now included a detailed description of the molecular analysis (see above).
Reviewer 2 Report
This case report is very interesting and well described. In my opinion, it would have been better to do a coprological investigation on the other individuals to verify if there were others positive. The adult parasites should have been counted and sexed. I advise authors to contact an English language reviewer.
Pag 1 line 27
has been reported
instead
have been reported
Pag 1 line 35-36
R: These signs are caused by verminous pneumonia provoked by migrating the first
stage larvae
instead of
These signs are caused by a verminous pneumonia provoked by migrating first stage larvae (L1)
Pag. 2 Line 80-83
One month before death, it had presented weakness, ataxia and paresis of the hindquarters. Albeit it was treated with a corticosteroid and antibiotic therapy, respiratory symptoms emerged, such as cough and dyspnea. A thoracic radio graph showed a diffuse nodular broncho-interstitial lung pattern (Figure 1). For the serious clinical conditions, it was decided to carry out euthanasia.
R: Why complete coprological analyzes, including the Baermann test, have not been done before?
Pag. 2 Line 88 page
Post-mortem examination revealed poor body conditions (Figure 2) and pale mucous
membranes. The major gross lesions were found in the lungs, where large, congested areas of
consolidation, particularly located in the basal lobes, were observed
instead of
Post-mortem examination revealed poor body conditions (Figure 2) and pale mucous membranes. The major gross lesions were found in lungs, where large congested areas of consolidation, particularly located in the basal lobes, were observed.
Pag. 3 Line 93-94
To check
Cardiomegaly was observed, with right ventricular hypertrophy associated with atrial dilatation
Instead of
Cardiomegaly was observed, with right ventricular hypertrophy associated to atrial dilatation
Pag. 3 Line 111
in the myocardium of the right ventricle, in the spleen, in the intestinal submucosa, and in portal 111
areas of the liver.
Instead of
in the myocardium of the right ventricle, in spleen, in intestinal submucosa and in portal areas of liver.
Pag 3 Line 113
To check
Extensive interstitial lymphoplasmacytic nephritis associated with marked fibrosis and scattered granulomas containing nematode larvae were seen in kidneys; larvae were also ob-114 served in many renal glomeruli (Figure 3c).
Instead of
Extensive interstitial lymphoplasmacytic nephritis associated to marked fibrosis and scattered granulomas containing nematode larvae were seen in kidneys; larvae were also ob-114 served in many renal glomeruli (Figure 3c).
Pag 4 Line 163-165
To the best of Authors’ knowledge, this is the first report of A. vasorum causing very
severe symptoms in an animal kept in a zoo in Italy, such to induce veterinarians to euthanize the animal, and it is the first report in V. zerda.
R: I would avoid saying it is the first case as it is obvious. I would emphasize the singularity of the event
Pag 4 line 165
Thus, the evidence that A. vasorum can cause mortality in or of captive animals is confirmed, since the clinical picture and the pathological findings observed in the fennec are suggestive that the parasite was presumably leading the animal to death
R: It seems a bit excessive to generalize and affirm “the evidence”. You only had one dead animal. As stated later the animal was probably not in good physical, immunological, etc. condition and therefore could be an exception.
Pag 4 Line 185
sometimes leading to captive animals’ death.
R: which other cases?
R: Being a parasitological study, why haven't the adult parasites been isolated, counted and sexed?
References
R species name in italics
Photos
it would have been nice to have a photo with adults too. Don't you have a picture of the larva at the highest magnification where it was possible to see better the tail?
Author Response
This case report is very interesting and well described. In my opinion, it would have been better to do a coprological investigation on the other individuals to verify if there were others positive. The adult parasites should have been counted and sexed. I advise authors to contact an English language reviewer.
- We thank the reviewer for all comments and suggestions. Actually, as indicated in lines 277-281, coprological examinations were carried out in the conspecific animals that live with the fennec of our paper. Animals were tested twice for the presence of this parasite via the Baerman technique. No other cases of infection were highlighted.
- We apologize but at the time of necropsy, after the finding of the first individuals, used for molecular identification, a thorough and complete search for other adults was not carried out, as we focused on the presence of L1 in the different tissues and organs, being the L1 disseminated presence in the different tissues the more relevant finding from a pathological point of view.
Pag 1 line 27
has been reported instead have been reported
- Done
Pag 1 line 35-36
These signs are caused by verminous pneumonia provoked by migrating the first stage larvae instead of These signs are caused by a verminous pneumonia provoked by migrating first stage larvae (L1)
- Done
Pag. 2 Line 80-83
One month before death, it had presented weakness, ataxia and paresis of the hindquarters. Albeit it was treated with a corticosteroid and antibiotic therapy, respiratory symptoms emerged, such as cough and dyspnea. A thoracic radio graph showed a diffuse nodular broncho-interstitial lung pattern (Figure 1). For the serious clinical conditions, it was decided to carry out euthanasia.
R: Why complete coprological analyzes, including the Baermann test, have not been done before?
- No parasitological tests were performed in life as a parasitic disease, such as angiostrongylosis, was not suspected as a differential diagnosis. In fact, the symptoms that emerged initially were characterized by neurological signs, as reported in the article. Respiratory signs emerged only two days before death. Following your helpful comment, we realized that we needed to describe more precisely what the clinical course of fennec was.
Pag. 2 Line 88 page
Post-mortem examination revealed poor body conditions (Figure 2) and pale mucous
membranes. The major gross lesions were found in the lungs, where large, congested areas of consolidation, particularly located in the basal lobes, were observed
instead of
Post-mortem examination revealed poor body conditions (Figure 2) and pale mucous membranes. The major gross lesions were found in lungs, where large congested areas of consolidation, particularly located in the basal lobes, were observed.
- Done
Pag. 3 Line 93-94
To check
Cardiomegaly was observed, with right ventricular hypertrophy associated with atrial dilatation
Instead of
Cardiomegaly was observed, with right ventricular hypertrophy associated to atrial dilatation
- Done
Pag. 3 Line 111
in the myocardium of the right ventricle, in the spleen, in the intestinal submucosa, and in portal areas of the liver.
Instead of
in the myocardium of the right ventricle, in spleen, in intestinal submucosa and in portal areas of liver.
- Done
Pag 3 Line 113
To check
Extensive interstitial lymphoplasmacytic nephritis associated with marked fibrosis and scattered granulomas containing nematode larvae were seen in kidneys; larvae were also observed in many renal glomeruli (Figure 3c).
Instead of
Extensive interstitial lymphoplasmacytic nephritis associated to marked fibrosis and scattered granulomas containing nematode larvae were seen in kidneys; larvae were also observed in many renal glomeruli (Figure 3c).
- Done
Pag 4 Line 163-165
To the best of Authors’ knowledge, this is the first report of A. vasorum causing very severe symptoms in an animal kept in a zoo in Italy, such to induce veterinarians to euthanize the animal, and it is the first report in V. zerda.
R: I would avoid saying it is the first case as it is obvious. I would emphasize the singularity of the event
- We think it is important to highlight this first report, both for the host species and for a zoo animal in particular, hence we preferred not to modify this point.
Pag 4 line 165
Thus, the evidence that A. vasorum can cause mortality in or of captive animals is confirmed, since the clinical picture and the pathological findings observed in the fennec are suggestive that the parasite was presumably leading the animal to death
R: It seems a bit excessive to generalize and affirm “the evidence”. You only had one dead animal. As stated later the animal was probably not in good physical, immunological, etc. condition and therefore could be an exception.
- We agree with the reviewer. We have modified the sentence.
Pag 4 Line 185
sometimes leading to captive animals’ death.
R: which other cases?
- Actually, we have reported in the introduction (line 102) some cases of fatal infection in captive animals. After the reviewer’s comment, in the revised manuscript we have now cited the articles present in the introduction also in the discussion.
R: Being a parasitological study, why haven't the adult parasites been isolated, counted and sexed?
- We apologize but at necropsy, after the finding of the first individuals, used for molecular identification, a thorough and complete search for other adults was not carried out, as we focused on the presence of L1 in the different tissues and organs, being the L1 disseminated presence in the different tissues the more relevant finding from a pathological point of view.
Reviewer 3 Report
Dear Authors, your manuscript is an interesting case report and I believe it is worth publishing after some corrections/modifications. Please see my specific comments below.
Line 11. Please delete “in captivity”
Line 12. Please replace the word symptoms with “signs” here and throughout the manuscript.
Line 15. Please replace “through” with “by”.
Line 18. Turn to plural: “parasitized gastropods”.
Line 18. Please add “or small vertebrates acting as paratenic hosts” right after “intermediate hosts”.
Line 28. Please delete “natural”.
Lines 30-32. Please organize better the mentioning of definitive hosts. It is a bit chaotic now as new animal species appear as the text progresses, while it seems like the parasite may not equally cause disease to all definitive hosts.
It would be better to not abbreviate the scientific name of the organisms when at the beginning of a sentence (multiple spots of the ms).
Line 35, Please delete “migrating”.
Line 35. Please add “eggs and” before “first”
Line 36. The primary reason for pulmonary hypertension (PH) in angiostrongylosis is not the presence of worms in the pulmonary vessels. Inflammatory lesions around the clades of the pulmonary artery cause PH too. This is also advocated by the fact that PH may persist despite the eradication of the parasites after treatment. Also, pulmonary hypertension is not the cause of thrombosis in this case, but may be the result of it. Please correct the pathogenesis in general. https://pubmed.ncbi.nlm.nih.gov/32847583/
https://pubmed.ncbi.nlm.nih.gov/32670858/
Line 40 Please add “aberrantly” after “caused by”.
Line 64. Please add “and small vertebrates paratenic hosts” before “entering”
Lines 80-83. I understand that parasitological and other laboratory examinations were not performed. Please mention this and the reason why such examinations did not take place, especially after the development of respiratory symptoms.
Line 80. “Condition” (singular)
Figure 3a. I believe there are also eggs visible in this section. Please indicate and mention it in the legend. https://pubmed.ncbi.nlm.nih.gov/19062192/
Line 133 Figure 4 shows a first-stage larva, not an adult female worm.
Line 158. Please replace “Specimen” with “animal”.
Line 175. Could also the consumption of paratenic hosts (e.g., rodents) be the way of infection of the animal? Maybe this way of infection could be added here as a possible scenario.
Lines 186-188. This sentence needs rephrasing as it is not conceptually correct.
Author Response
Dear Authors, your manuscript is an interesting case report and I believe it is worth publishing after some corrections/modifications. Please see my specific comments below.
Line 11. Please delete “in captivity”
- Done
Line 12. Please replace the word symptoms with “signs” here and throughout the manuscript.
- Done
Line 15. Please replace “through” with “by”.
- Done
Line 18. Turn to plural: “parasitized gastropods”.
- Done
Line 18. Please add “or small vertebrates acting as paratenic hosts” right after “intermediate hosts”.
- Done
Line 28. Please delete “natural”.
- Done
Lines 30-32. Please organize better the mentioning of definitive hosts. It is a bit chaotic now as new animal species appear as the text progresses, while it seems like the parasite may not equally cause disease to all definitive hosts.
- Done
It would be better to not abbreviate the scientific name of the organisms when at the beginning of a sentence (multiple spots of the ms).
- Done
Line 35, Please delete “migrating”.
- Done
Line 35. Please add “eggs and” before “first”
- Done
Line 36. The primary reason for pulmonary hypertension (PH) in angiostrongylosis is not the presence of worms in the pulmonary vessels. Inflammatory lesions around the clades of the pulmonary artery cause PH too. This is also advocated by the fact that PH may persist despite the eradication of the parasites after treatment. Also, pulmonary hypertension is not the cause of thrombosis in this case, but may be the result of it. Please correct the pathogenesis in general. https://pubmed.ncbi.nlm.nih.gov/32847583/
https://pubmed.ncbi.nlm.nih.gov/32670858/
- We thank the reviewer for the comment and we have proceeded to modify the pathogenesis, describing it more precisely. We have also cited one of the two papers that he/she indicated to us, inserting the reference.
Line 40 Please add “aberrantly” after “caused by”.
- Done
Line 64. Please add “and small vertebrates paratenic hosts” before “entering”
- Done
Lines 80-83. I understand that parasitological and other laboratory examinations were not performed. Please mention this and the reason why such examinations did not take place, especially after the development of respiratory symptoms.
- As already explain to reviewer 2 , no parasitological tests were performed in life as a parasitic disease, such as angiostrongylosis, was not suspected as a differential diagnosis. In fact, the symptoms that emerged initially were characterized by neurological signs, as reported in the article. Respiratory signs emerged only two days before death. Following your helpful comment, we realized that we needed to better describe the clinical evolution of disease. As suggested we also mentioned in the manuscript the reason why parasitological tests were not performed. We also did not mention that a blood count was performed which revealed a mild lymphocytosis. Therefore we have added this exam in the case presentation.
Line 80. “Condition” (singular)
- Done
Figure 3a. I believe there are also eggs visible in this section. Please indicate and mention it in the legend. https://pubmed.ncbi.nlm.nih.gov/19062192/
- We thank the reviewer for the suggestion. Indeed, we did not indicate the eggs present in the histological image. Accordingly, we changed the panel.
Line 133 Figure 4 shows a first-stage larva, not an adult female worm.
- We changed the wrong quote by inserting another sentence to show larvae isolated from impression smears of the lungs.
Line 158. Please replace “Specimen” with “animal”.
- Done
Line 175. Could also the consumption of paratenic hosts (e.g., rodents) be the way of infection of the animal? Maybe this way of infection could be added here as a possible scenario.
We thank the reviewer for his/her comment. However, we think that slugs and snails entering zoo enclosures from the outside is a common occurrence. Moreover the role of possible paratenic hosts in A. vasorum life cycle is not so sure. Hence, we preferred not to modify the manuscript on this point.
Lines 186-188. This sentence needs rephrasing as it is not conceptually correct.
- We changed the sentence.
References R species name in italics.
- Done
Photos
it would have been nice to have a photo with adults too. Don't you have a picture of the larva at the highest magnification where it was possible to see better the tail?
- As requested, we have subsequently highlighted the details of the larva's tail. We apologise, but we didn’t take any picture of the adult parasites, before submitting them to molecular analysis. However, as the L1 is the diagnostic stage of the parasite at coprological examination, we think that L1 photo is probably more relevant.
Reviewer 4 Report
TITLE: Systemic infection by Angiostrongylus vasorum in a fennec (Vulpes zerda) in an Italian zoological garden
CODE: pathogens -1846520
The authors reported an interesting case of a Angiostrongylus vasorum systemic infection in a captive fennec (Vulpes zerda) in a zoo of Italy. However, several details must be addressed before publication. See corrections below.
Case presentation
- Change paresis of hindquarters for “paraparesis” (lines 80-81)
- Specify which corticosterioids and antibiotics were used, frequency and doses (lines 81-82)
- Change “for the serious clinical conditions..” for “Due to serious clinical condition and not improvement with the treatment protocol , it was decided….” or similar (lines 83-84)
- Change “basal lobes” for “caudal lobes” (line 90)
- Please include macroscopic and microscopic findings of the thoracolumbar or lumbosacral spinal cord that could explain the paraparesis and/or pelvic limb ataxia (lines 109)
Disussion
- Include, if possible, epidemiological aspects of the prevalence of A.vasorum in studies of the region of Rome that include prevalence in gastropods or clinical cases in dogs.
Author Response
The authors reported an interesting case of a Angiostrongylus vasorum systemic infection in a captive fennec (Vulpes zerda) in a zoo of Italy. However, several details must be addressed before publication. See corrections below.
Case presentation
Change paresis of hindquarters for “paraparesis” (lines 80-81)
- Done
Specify which corticosteroids and antibiotics were used, frequency and doses (lines 81-82)
- We have rightly specified, as required, which cortisone and antibiotics were used, also adding the times and frequency of administration.
Change “for the serious clinical conditions..” for “Due to serious clinical condition and not improvement with the treatment protocol , it was decided….” or similar (lines 83-84)
- Done
Change “basal lobes” for “caudal lobes” (line 90)
- Done
Please include macroscopic and microscopic findings of the thoracolumbar or lumbosacral spinal cord that could explain the paraparesis and/or pelvic limb ataxia (lines 109)
- Macroscopic examination did not reveal any significant lesions of the spinal cord. The thoracolumbar tract was taken for histological examination, but initially the sections examined did not show major lesions. Following your comment, we decided to make further cuts of the samples from which angiostrongylosis lesions emerged. In fact, we found A. vasorum larvae associated with small lymphocytic infiltrates. We have therefore also included an additional histological image in figure 3 (Figure 3e). Therefore, we thank the reviewer for his/her helpful suggestion.
Discussion
Include, if possible, epidemiological aspects of the prevalence of A. vasorum in studies of the region of Rome that include prevalence in gastropods or clinical cases in dogs.
- In the last years many papers were published on A. vasorum in central Italy, but prevalence data from the region of Rome are lacking. Moreover, since this is a case report of a very peculiar finding, we focused on the case and we preferred not to load the paper with a long epidemiological discussion or a long list of clinical cases in dogs, to make the paper well focused on the point.
Round 2
Reviewer 1 Report
Minor corrections:
line 85: gastropod
lines 222, 260, 262: Latin names shall be in italic font
Reviewer 4 Report
The authors made the suggested modifications. The article may be published in its current form. Only a minor suggestion:
Figure 1: Change "Chest X-ray..." for "Thoracic radiograph..."